# A New Nomogram-Based Prediction Model for Postoperative Outcome after Sigmoid Resection for Diverticular Disease

**DOI:** 10.3390/medicina59061083

**Published:** 2023-06-04

**Authors:** Sascha Vaghiri, Sarah Krieg, Dimitrios Prassas, Sven Heiko Loosen, Christoph Roderburg, Tom Luedde, Wolfram Trudo Knoefel, Andreas Krieg

**Affiliations:** 1Department of Surgery (A), Heinrich-Heine-University, University Hospital Duesseldorf, 40225 Duesseldorf, Germany; sascha.vaghiri@med.uni-duesseldorf.de (S.V.); dimitrios.prassas@med.uni-duesseldorf.de (D.P.); 2Clinic for Gastroenterology, Hepatology and Infectious Diseases, Heinrich-Heine-University, University Hospital Duesseldorf, 40225 Duesseldorf, Germany; sarah.krieg@med.uni-duesseldorf.de (S.K.); sven.loosen@med.uni-duesseldorf.de (S.H.L.); christoph.roderburg@med.uni-duesseldorf.de (C.R.); tom.luedde@med.uni-duesseldorf.de (T.L.)

**Keywords:** morbidity and mortality, postoperative length of stay, nomogram-based prediction, sigmoid diverticulitis

## Abstract

*Background and Objectives:* Sigmoid resection still bears a considerable risk of complications. The primary aim was to evaluate and incorporate influencing factors of adverse perioperative outcomes following sigmoid resection into a nomogram-based prediction model. *Materials and Methods:* Patients from a prospectively maintained database (2004–2022) who underwent either elective or emergency sigmoidectomy for diverticular disease were enrolled. A multivariate logistic regression model was constructed to identify patient-specific, disease-related, or surgical factors and preoperative laboratory results that may predict postoperative outcome. *Results:* Overall morbidity and mortality rates were 41.3% and 3.55%, respectively, in 282 included patients. Logistic regression analysis revealed preoperative hemoglobin levels (*p* = 0.042), ASA classification (*p* = 0.040), type of surgical access (*p* = 0.014), and operative time (*p* = 0.049) as significant predictors of an eventful postoperative course and enabled the establishment of a dynamic nomogram. Postoperative length of hospital stay was influenced by low preoperative hemoglobin (*p* = 0.018), ASA class 4 (*p* = 0.002), immunosuppression (*p* = 0.010), emergency intervention (*p* = 0.024), and operative time (*p* = 0.010). *Conclusions:* A nomogram-based scoring tool will help stratify risk and reduce preventable complications.

## 1. Introduction

Sigmoid diverticulitis has become an increasingly common disease in recent decades. Extensive epidemiological surveillance shows an annual increase in the hospitalization rate associated with acute colonic diverticulitis in the U.S. and in European countries requiring urgent surgical intervention [1,2,3]. In principle, the broad spectrum of diverticular disease ranges from symptomatic uncomplicated diverticular disease (SUDD) to acute uncomplicated subtypes with peridiverticular phlegmon and complicated sequelae such as covered and free perforation, stenosis, or fistula, with therapy usually depending on the patient and disease stage [4]. Frank perforation with generalized peritonitis or unsuccessful initial conservative treatment requires immediate or urgent surgical resection, whereas elective sigmoidectomy is reserved for patients with severe or recurrent episodes after complete symptom relief, taking into account complicated chronic manifestations, relevant comorbidities, and the individual’s quality of life [5]. Overall morbidity and mortality after elective sigmoid resection have been reported to be 25–28% and 0.1–1.2%, respectively [6,7], in contrast to emergency sigmoidectomy, where morbidity rates of 36–63% and postoperative mortality rates of 6.9–12% have been observed [8,9]. These numbers highlight the importance of identifying factors significantly associated with morbidity and mortality, in an attempt to reduce surgery-related complications. To date, few studies have exclusively examined factors associated with unfavorable outcomes after surgical treatment of sigmoid diverticular disease [6,7,10,11].

Recently, risk-adjusted scoring systems for colorectal emergency and elective surgery have been introduced into clinical practice, including the Portsmouth physiological and operative severity score for the enumeration of mortality and morbidity (P-POSSUM) [12], the National Emergency Laparotomy Audit (NELA) score [13], and the American College of Surgeons National Surgical Quality Improvement Program (ACS-NSQIP) calculator [14], with varying performance levels [15,16]. However, their validity and accuracy for sigmoid diverticulitis must be questioned, because colorectal cancer, inflammatory bowel disease, and diverticular disease represent distinct entities with different underlying pathophysiologic mechanisms [17].

A specific risk-scoring tool to predict a complicated postoperative course after sigmoid resection due to diverticular disease is still scarce despite its immense clinical relevance in daily practice. Furthermore, factors influencing postoperative hospital stay in this area have not been described to date. Therefore, the primary aim of this study was to investigate in detail the influence of patient-, disease-, and surgery-related factors, as well as laboratory findings on the postoperative course in patients undergoing sigmoid resection for diverticular disease, thus enabling appropriate risk stratification by a generated nomogram.

## 2. Materials and Methods

This study was based on our prospectively maintained clinical database of all patients admitted to the Department of Surgery (A) of the University Hospital Duesseldorf between January 2004 and September 2022 for acute sigmoid diverticulitis. Relevant information was retrieved by critical review of clinical and outpatient medical records and radiographic results. Our analysis included all patients who underwent sigmoidectomy, either on initial hospital admission or on a second stay after complete relief of diverticulitis-triggered symptoms.

Clinical data were systematically collected, including: (1) demographic data (i.e., age, sex, body mass index (BMI), and American Society of Anesthesiologists (ASA) classification) and pre-existing comorbidities (e.g., cardiovascular disease, metabolic disease, or immunosuppression), number and disease stage of previous attacks, radiological assessment, and laboratory parameters (including c-reactive protein (CRP), leukocytes, hemoglobin, and thrombocytes); (2) treatment strategy, failure rates of initial medical treatment, and surgical approach or intraoperative course (e.g., conversion rate or stoma creation); (3) perioperative surgical complications (e.g., wound infection, burst abdomen, anastomotic leak or stenosis, gastrointestinal leak other than anastomotic leak, postoperative ileus, or intraabdominal abscess formation) or medical complications (e.g., sepsis, pneumonia, renal failure, or cardiovascular events), and re-operation or intervention; and (4) in-hospital mortality, and postoperative length of hospital stay.

On admission, all patients with clinical suspicion of acute sigmoid diverticulitis were evaluated interdisciplinary and underwent CT imaging to confirm the diagnosis. The disease stage was finally defined using the modified Hinchey classification [18]. Briefly, disease severity was classified as uncomplicated for modified Hinchey stages 0 and IA, and as complicated for modified Hinchey stages IB, II, III, and IV, or chronic inflammatory complications such as stenosis, stricture, fistula, and hemorrhage. The number of episodes was defined as the number of any new acute attacks after successful inpatient or outpatient medical treatment (depending on disease severity and clinical symptoms) of radiologically-confirmed sigmoid diverticular disease. Failure of non-operative treatment was determined by persistence or worsening of diverticulitis-associated symptoms and objective parameters within 72 h of initiation of treatment [19]. Either insertion of a percutaneous abscess drain or primary or secondary emergency surgery was declared to be an emergency intervention. The Clavien–Dindo classification [20] was applied to classify perioperative morbidity, whereby we defined all complications starting from a Clavien–Dindo grade I complication as an eventful postoperative course. Upon presentation to our central emergency department, all patients with a suspected diagnosis of diverticulitis were evaluated by an interdisciplinary team of general/visceral surgeons and internists. In general, at our university hospital, all patients with an indication for surgery or interventional therapy to treat sigmoid diverticular disease are transferred from the central emergency department to the surgical department.

Treatment in our clinic was carried out according to the recommendations of the German Society of Gastroenterology, Digestive and Metabolic Diseases (DGVS) and the German Society for General and Visceral Surgery (DGAV) [21]. According to these recommendations, clinically stable patients with complicated diverticular disease are to be treated as inpatients receiving antibiotic therapy. In the presence of free perforation, emergency surgery was performed, which included median laparotomy and sigmoid resection with either creation of a terminal stoma or primary anastomosis with a protective stoma in relatively stable patients. The decision to perform either a Hartmann’s resection or a primary anastomosis with or without a protective stoma was significantly influenced by intraoperative findings, the patient’s general health, and the surgeon’s individual preference. In cases with severe purulent or fecal peritonitis associated with hemodynamic instability, sigmoid resection with blind closure followed by staged peritoneal lavage (SPL) and stoma creation was performed for damage control. In covered perforations with larger abscess collections, primary percutaneous abscess drainage was performed. After successful medical treatment, patients were considered for either early elective or delayed elective resection at 4–6 weeks, depending on the degree of disease burden, significant comorbidities, or impaired quality of life. In these cases, we performed laparoscopic sigmoid resection with the routine use of four ports, mobilization from lateral to medial, and low tie ligation.

### 2.1. Outcome of Interest

In this study, we aimed to elucidate factors associated with postoperative outcome and length of hospital stay as functions of patient-specific characteristics and perioperative course, respectively, by performing a binary logistic regression analysis in a large cohort of patients undergoing elective or emergency sigmoidectomy.

### 2.2. Statistical Analysis

Statistical analysis was performed using G*Power [22], R software package (version R4.1.1, R Foundation for statistical computing) [23] and the readxl [24], psych [25], MICE [26], bootStepAIC [27], Resource Selection [28], rms [29], DynNom [30], and MASS [31] packages. Continuous variables were presented as median with standard deviation (SD). For descriptive data analysis, continuous variables were tested using the Mann–Whitney U test, and categorical variables were analyzed using Fisher’s exact test or the chi-square test. The multivariate imputation by chained equations (MICE) algorithm was applied to impute missing values in our dataset. Note that only variables in which < 20% of the data were missing were used. A binary logistic regression model was used when the dependent variable included two categories. The clinical selection model was constructed as recently described [32]. Briefly, we first screened for independent variables based on backward stepwise selection, where the model with the lowest Akaike information criterion (AIC) is selected and implemented a bootstrap procedure by resampling 100 times to investigate the variability of model selection. Next, we ran the logistic regression model suggested by the bootstrap method. Finally, the independent predictors were used to construct a nomogram. The discriminatory power of the model was evaluated using the C statistic. A C-index of 1 means that the model predicts the outcome perfectly, a value above 0.8 or above 0.7 indicates a strong or good model, respectively, and a value of 0.5 means that the model does not predict better than chance. The Brier score was used to measure the accuracy of probabilistic predictions. A Brier score can take any value between 0 and 1, where 0 is the best achievable value and 1 is the worst achievable value. The lower the Brier score, the more accurate the prediction. Goodness of fit (GOF) was assessed using the Hosmer–Lemeshow test, with *p*-values < 0.05 indicating poor fit. Due to the small sample size, we internally validated the reproducibility of our nomogram using the bootstrap method by drawing a new sample 100 times and assessing the calibration curves. For a dependent continuous variable, we performed a linear regression model with stepwise backward selection. In all analyses, a *p*-value < 0.05 indicated statistical significance. The post hoc statistical significance of the logistic regression analysis of the eventful postoperative course according to the primary surgical approach and the multivariate linear regression analysis of the predictors of length of stay were 96.40% and 99.98%, respectively, at a significance level of 0.05.

## 3. Results

### 3.1. Patient Characteristics and Operative Data

A total of 282 patients (146 female/136 male) with a median age of 59.0 ± 13.35 (range 26–94) years were included for the final analysis (Table 1). Unfortunately, due to missing or incomplete records, we were unable to collect a history of recent bowel surgery for all patients.

In about half of the patients, arterial hypertension was prevalent (47.87%), and 14.18% were receiving immunosuppressive therapy. The immunosuppression group consisted of twenty patients with autoimmune disease (including four cases of chronic inflammatory bowel disease) taking oral corticosteroids, fifteen patients with renal transplantation, three patients with human immunodeficiency virus (HIV), one patient with combined renal and cardiac transplantation, and one patient with metastatic cancer receiving chemotherapy. All patients with autoimmune disease were taking corticosteroids of some subtype for at least four weeks prior to surgery. In addition to corticosteroids, non-corticosteroid concomitant medications (e.g., methotrexate, cyclosporine, azathioprine, cyclophosphamide, and rituximab) were recorded in eight patients. In transplant patients, perioperative immunosuppression was maintained with tacrolimus, in addition to corticosteroids and mycophenolic acid when indicated. According to the World Health Organization (WHO) definition of anemia [33], 64 included patients (29 male/35 female) had preoperative anemia (22.69%). Of note, none of the patients enrolled had a history of previous infectious colitis, and the majority of patients presented to our department with their first episode of diverticular disease (56.38%), while recurrent attacks were observed in almost 44%. Complicated diverticular disease was the most common reason for admission (76.60%) and required urgent surgical resection in 34.04% of patients (*n* = 96). Interestingly, initial conservative treatment, including percutaneous abscess drainage, was unsuccessful in 9.93% of cases because symptoms persisted or even worsened. The median time interval between disease onset or hospitalization and surgery was 9.0 ± 31.79 days. The surgical approach was equally distributed, with 49.65% of patients (*n* = 140) undergoing laparoscopic sigmoidectomy and 50.35% (*n* = 142) open sigmoidectomy (Table 2).

The conversion rate in the laparoscopic group was 20.71% (29 out of 140 patients). The main reasons for conversion were severe inflammatory adhesions (*n* = 18), conglomerates (*n* = 8), and bleeding (*n* = 3). Hartmann resection was performed in 82 patients (29.08%), while primary anastomosis with and without protective ostomy was performed in 9.22% and 61.70%, respectively. The median operative time was 245.0 ± 82.86 min for the entire cohort.

### 3.2. Postoperative Outcome

The overall morbidity rate in the studied cohort was 41.13%. Postoperative surgical and medical complications are summarized in Table 3. A total of 89 (31.56%) wound infections were identified as the most common postoperative adverse event. Burst abdomen and fascial insufficiency were observed in 15 patients (5.32%). Paralytic or mechanical bowel obstruction was found in 5.32%. In addition, anastomotic leakage was noted in 5.75% of cases. Medical complications were attributed to sepsis (4.26%), renal failure (3.55%), cardiovascular events (3.19%), and pneumonia (2.13%). A re-intervention was required in 15.25%. A total of 10 patients died during hospital stay, representing a mortality rate of 3.55%.

### 3.3. Factors Associated with Occurrence of Morbidities and Prolonged Hospital Stay

First, we compared patient characteristics for uneventful and eventful (defined as at least one grade I complication according to the Clavien–Dindo classification) postoperative courses (Table 4).

This indicated that older age, diabetes mellitus, arterial hypertension, higher ASA classification, immunosuppression, lower hemoglobin levels, complicated current episode, first diverticular episode, emergency intervention, shorter time interval between current episode and surgery, open surgical access, and Hartmann resection were associated with a complicated postoperative course. In order to test the hypothesis that these factors were suitable for predicting a complicated postoperative course, we next performed a multivariate binary logistic regression analysis. For this purpose, we included all clinically relevant variables listed in Table 4, imputed all missing values, and used stepwise backward selection and bootstrap resampling to select the independent variables. By using this approach, we were able to identify leukocyte count, hemoglobin levels, ASA classification, diabetes mellitus, time between current episode and surgical intervention, surgical access, and surgical duration as factors that were associated with a complicated postoperative course (Table 5).

Based on this final model, we constructed a nomogram for predicting the likelihood of a complicated postoperative course (Figure 1A). In this nomogram, a perpendicular line is drawn from each of the predictors to the point axis. The points obtained in this way are then added to give the total points, from which a vertical line is finally drawn to the “probability (eventful postoperative course)” axis, which, in turn, represents the probability of an eventful postoperative course.

Notably, a C-index of 0.736 and a Brier score of 0.200 indicated a good model with accurate predictive power. Furthermore, the Hosmer–Lemeshow test verified the GOF (*p* = 0.2603) and the calibration curve was nearly in congruence with the ideal line, suggesting a well-calibrated model (Figure 1B).

In addition, we created a dynamic nomogram from the model; the nomogram provides a helpful web-based tool for daily routine use and can be used to individually calculate the risk for an eventful course. The example in Figure 2 shows the risk for a patient with 8000 leukocytes/µL, a hemoglobin of 15 mg/dl, no diabetes mellitus, ASA classification of 1, 2 days from current episode to surgery, use of laparoscopic surgery, and an operation time of 109 min. The graphical and numerical summary calculates only a 10.9 percent risk of a postoperative complication for this patient.

Next, we sought to identify factors that influence the length of hospital stay after sigmoid resection. Therefore, we compared patient characteristics with the length of hospital stay (Table 6). Patients with diabetes mellitus, arterial hypertension, chronic renal insufficiency, high ASA classification, immunosuppression, a single diverticulitis episode, increasing severity of diverticulitis, complicated current episode, emergency intervention such as interventional abscess drainage or emergency surgery, failure of conservative therapy, open surgical approach, and Hartmann procedure showed a longer postoperative hospital stay.

Using a stepwise backward linear regression analysis which included the variables summarized in Table 6, we built a model to identify independent factors predicting length of hospital stay. This model confirmed that hospital stay was prolonged in patients with limited health status according to the ASA classification, low preoperative hemoglobin, immunosuppression, emergency intervention, and for whom a longer operation time was recorded (Table 7).

## 4. Discussion

The primary objective of our study was to identify factors influencing postoperative morbidity and length of hospital stay after sigmoid resection for diverticular disease. For this purpose, we included 282 patients from our prospectively maintained database who underwent either elective or emergency surgery for sigmoid diverticulitis at our hospital. It is important to note that the morbidity and mortality rates in our cohort after sigmoidectomy were 41.3% and 3.55%, respectively, which is comparable to previously reported data [9]. From the combined parameters of preoperative leukocyte count and hemoglobin, ASA classification, surgical procedure (laparoscopic versus open), diabetes mellitus, time between current episode and surgery, and duration of surgery, we were able to derive a model that served as the basis for creating a nomogram that can be used as a clinical tool for risk stratification before surgical therapy for sigmoid diverticulitis. Furthermore, our analysis showed that protracted postoperative hospital stay was associated with high ASA classification, low preoperative hemoglobin, suppressed immune system, emergency intervention, and duration of the surgical procedure. The novelty of our work here is the creation of a practical, easy-to-use nomogram that can be conveniently applied in daily clinical practice. In addition, we provide predictive factors for prolonged hospital stay that, to our knowledge, have not been described previously.

Sigmoid diverticular disease is a common condition, and surgical treatment remains the mainstay of therapy, both in emergency cases and, in some cases, during the inflammation-free interval as an elective procedure [5,19]. Despite recent advances in diagnostic methods and inpatient care over the past few decades, overall morbidity and mortality rates after sigmoid resection are unfortunately still very high, even for elective procedures [8,9]. These risk factors for surgical complications are partly due to patient- and disease-specific factors, but are also related to the surgical procedure itself, to varying degrees [34]. Therefore, it is crucial to identify patient- and disease-specific factors that are significantly associated with adverse postoperative events in order to be able to reduce morbidity and mortality rates in the future. Furthermore, with this knowledge, a more-tailored therapeutic approach would be possible for each patient. Many relevant factors associated with increased morbidity and mortality in colorectal surgery have been evaluated in the past [35,36]. However, the wide range of surgical indications, from malignancies to inflammatory and functional disorders, may affect their application and validity in the context of sigmoid resection for diverticular disease. To date, few studies have specifically examined the impact of influencing factors on perioperative outcomes in diverticular disease. Pessaux et al. [6] were the first to examine risk factors for an eventful postoperative course after elective open sigmoid resection, with a sample of 582 patients. They found that lack of antibiotic coverage was the only significant determinant of abdominal complications, whereas chronic lung disease or liver cirrhosis accounted for extra-abdominal morbidity in multivariate analysis. The studies published by Kirchhoff et al. [10] and Silva-Velazco et al. [7] analyzed predictive factors after laparoscopically performed elective sigmoidectomy. Thus, Kirchhoff and colleagues [10] revealed that anemia (Hb ≤ 12 g/L), heart failure, previous myocardial infarction, surgeon’s experience, and male gender were predictive factors for postoperative complications, while an age older than 75 years turned out to be the only significant factor for intraoperative complications. On the other hand, the intention-to-treat multivariate regression analysis by Silva-Velazco et al. [7] indicated ASA classification 3, conversion to open surgery, and linear stapler versus knife transection technique as significant factors of postoperative morbidity. Another study by Skala et al. [37] revealed that ASA classification ≥3, colonic ischemia, and stoma creation significantly correlated with postoperative morbidity in a cohort that underwent emergency colorectal surgery, including 28% diverticulitis cases.

Consistent with these results, we also identified ASA classification and decreased hemoglobin levels as predictive factors for a complicated postoperative course. However, in addition, the type of surgical procedure (laparoscopic versus open), surgical time, incidence of diabetes mellitus, and the time between the last disease episode and surgery also determined postoperative morbidity in our study. In contrast to other publications analyzing either elective or emergency cases, we believe that for the present study a very notable strength is the representative cross-section of patients who underwent procedures including both laparoscopic and open emergency (or elective) sigmoidectomy for diverticular disease in a surgical department. The results presented have several useful implications for daily clinical practice. Special attention should be paid to preoperative laboratory markers such as hemoglobin and leukocytes. An increase in hemoglobin levels, preferably above 13.5 g/dL, is recommended before surgery as preoperative anemia significantly affects adverse outcomes after colorectal surgery [38]. When possible, a minimally invasive procedure should be the first approach to reduce postoperative morbidity and hospital stay. Indeed, laparoscopic sigmoidectomy has been shown to be consistently superior to open surgery in terms of shorter hospital stay, faster recovery, lower morbidity, and better quality of life [39,40]. In addition, the duration of surgery must be kept as short as possible. A longer duration of surgery has been shown to be associated with more perioperative complications [41]. In many studies, elevated blood glucose levels are directly linked to adverse perioperative events [42,43] and we have also been able to identify diabetes mellitus as a predictor of an eventful postoperative outcome, although we cannot provide information on the individual blood glucose levels of patients during the perioperative course. Consequently, careful optimization of blood glucose levels should therefore also be recommended during sigmoid resection for diverticular disease. While chronic renal insufficiency and impaired renal function measured by glomerular filtration rate (GFR) were independently associated with increased complications after abdominal surgery in two large observational studies [44,45], our results did not demonstrate a significant impact of renal insufficiency and reduced GFR rates on postoperative outcomes, possibly due to the small number of included patients with chronic kidney disease.

Furthermore, our data show that the time interval between disease onset and surgical therapy has the potential to negatively influence the perioperative course. Therefore, especially in emergency cases or in patients in whom primary medical treatment has failed, a timely and critical reassessment of the current clinical condition after 72 h from the start of conservative therapy, as suggested by Hanna et al. [19], is crucial and should prompt surgical resection if warranted. Simultaneously, elective sigmoidectomy should be scheduled in the inflammation-free interval 4–6 weeks after complete symptom amelioration in complicated disease stages (e.g., diverticular abscess) [32]. Of note, in our nomogram, decreasing leukocyte levels were inversely associated with the occurrence of perioperative morbidity. One possible explanation is that we included 40 patients with immunosuppression in our analysis, who had a morbidity rate of 62.5% with a median preoperative leukocyte count of 10.0 ± 6.18 (×1000/µL).

Our study has some limitations related to its retrospective design. Selection bias is an important confounding factor in this context. In addition, data collection over a long period of time involving many surgeons with different types of experience in surgical management and the progress of medical care during this period could influence not only intraoperative course but also surgery-related complications.

## 5. Conclusions

Sigmoid resection for diverticular disease is associated with high perioperative morbidity. By identifying factors associated with an unfavorable postoperative course and prolonged hospital stay, appropriate risk stratification and optimization of variables susceptible to influence in patients undergoing elective or emergency sigmoidectomy could allow reductions of operative complications and in the length of hospital stay. The proposed nomogram, derived from a representative study cohort, showed accurate predictive power and allowed us to create a dynamic nomogram for web application that should be validated in future prospective studies.

## Figures and Tables

**Figure 1 medicina-59-01083-f001:**
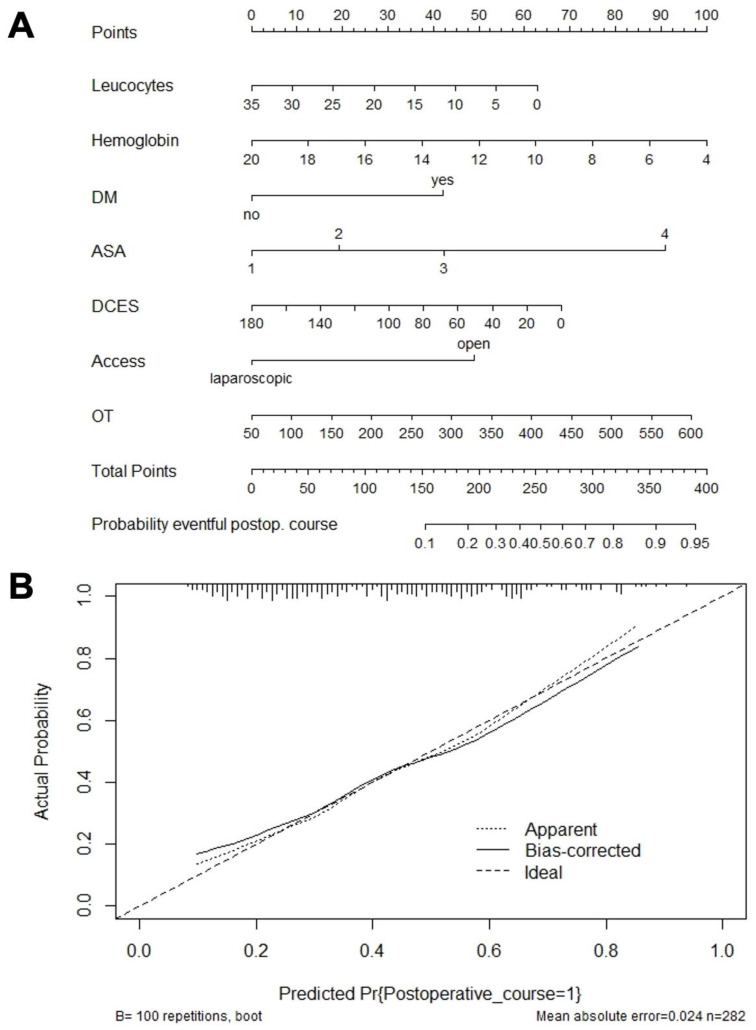
Nomogram for predicting the probability of an eventful postoperative course in patients undergoing sigmoid resection for diverticular disease. The nomogram (**A**) was constructed by multivariate regression analysis, and the calibration curve (**B**) revealed a well calibrated model. Leucocytes: number ×1000/µL; Hemoglobin: g/dl. Abbreviations: DM, Diabetes mellitus; DCES, Duration from current episode to surgery in days; OT, Operation time.

**Figure 2 medicina-59-01083-f002:**
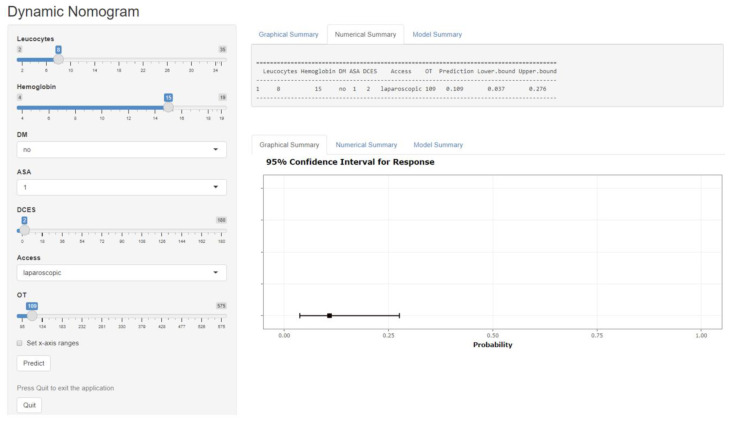
Dynamic nomogram for predicting the probability of an eventful postoperative course in patients undergoing sigmoid resection for diverticular disease. A helpful web-based tool for daily routine allows the user to calculate the risk for an individual eventful postoperative course.

**Table 1 medicina-59-01083-t001:** Patient characteristics and clinical status.

	All Patients *n* = 282
**Age (median ± SD)**	59.0 ± 13.35
**Female/Male ratio (*n*; %)**	146/136 (51.77/48.23)
**BMI (kg/m^2^) [median ± SD]**	26.57 ± 5.59
**ASA classification (*n*; %)**	
1	35 (12.41)
2	118(41.84)
3	74 (26.24)
4	10 (3.55)
NA	45 (15.96)
**Comorbidities (*n*; %)**	
Diabetes	20 (7.09)
Arterial hypertension	135 (47.87)
Chronic renal insufficiency	32 (11.35)
Immunosuppression	40 (14.18)
**Preoperative laboratory data**	
CRP (mg/dL) [median ± SD]	3.75 ± 10.07
Leucocytes (×1000/µL) [median ± SD]	10.20 ± 5.53
Hemoglobin (g/dl) [median ± SD]	13.50 ± 2.16
Thrombocytes (×1000/µL) [median ± SD]	264.0 ± 115.88
Number of episodes (*n*; %)	
1	159 (56.38)
2	51 (18.09)
3	52 (18.44)
≥4	20 (7.09)
**Modified Hinchey classification (*n*; %)**	
0	8 (2.84)
IA *	62 (21.99)
IB	105 (37.23)
II	27 (9.57)
III	69 (24.47)
IV	11 (3.90)
Complicated current episode (*n*; %)	216 (76.60)
Emergency intervention (*n*; %)	106 (37.59)
Percutaneous abscess drainage (*n*; %)	17 (6.03)
Failure initial medical therapy (*n*; %)	28 (9.93)
Emergency surgery (*n*; %)	96 (34.04)
Duration current episode to surgery (days) [median ± SD]	9.0 ± 31.79

BMI: Body mass index, ASA (classification): American Society of Anesthesiologists, CRP: C-reactive protein, NA: not available, * 4 patients with fistula formation.

**Table 2 medicina-59-01083-t002:** Intraoperative data.

	All Patients *n* = 282
Primary laparoscopic (*n*; %)	140 (49.65)
Primary open (*n*; %)	142 (50.35)
Conversion to open procedure (*n*; %)	29/140 (20.71)
Hartmann’s resection (*n*; %)	82 (29.08)
Primary anastomosis without ostomy (*n*; %)	174 (61.70)
Primary anastomosis with protective ostomy (*n*; %)	26 (9.22)
Operative time (min) [median ± SD]	245.0 ± 82.86

**Table 3 medicina-59-01083-t003:** Postoperative course.

	All Patients *n* = 282
Overall postoperative morbidity (*n*; %)	116 (41.13)
**Surgical complications (** ** *n* ** **; %)**	
Wound infection	89 (31.56)
Burst abdomen	15 (5.32)
Anastomotic leak *	10 (5.75)
Anastomotic stenosis	0 (0)
Ileus (mechanic/paralytic)	15 (5.32)
Intra-abdominal abscess	2 (0.71)
Remote GI leak	6 (2.13)
**Medical complications (** ** *n* ** **; %)**	
Sepsis	12 (4.26)
Pneumonia	6 (2.13)
Renal failure	10 (3.55)
Cardiovascular events	9 (3.19)
**Clavien–Dindo Classification (** ** *n* ** **; %)**	
0	166 (58.87)
I	10 (3.55)
IIa	35 (12.41)
IIb	23 (8.16)
IIIa	30 (10.64)
IIIb	6 (2.13)
IV	2 (0.71)
V	10 (3.55)
Re-operation (*n*; %)	43 (15.25)
Hospital stay (days) [median ± SD]	11.0 ± 15.81

GI leak: gastrointestinal leak, * *n* = 174 patients with anastomosis.

**Table 4 medicina-59-01083-t004:** Characteristics of cases with and without postoperative morbidity.

	No Postoperative Morbidity *n* = 166	Postoperative Morbidity *n* = 116	*p*-Value
Female/Male ratio	81/85 (48.80/51.20)	65/51 (56.03/43.97)	0.275
Age (years) [median ± SD]	56.0 ± 12.81	61.5 ± 13.47	<0.0001
BMI (kg/m²) [median ± SD]	26.87 ± 5.45	26.45 ± 5.81	0.382
Diabetes (*n*; %)	7 (4.22)	13 (11.21)	0.032
Arterial hypertension (*n*; %)	66 (39.76)	69 (59.48)	0.001
Chronic renal insufficiency (*n*; %)	15 (9.04)	17 (14.66)	0.181
**ASA classification (*n*; %)**			<0.0001
1	28 (16.87)	7 (6.03)	
2	79 (47.59)	39 (33.62)	
3	33 (19.88)	41 (35.34)	
4	3 (1.81)	7 (6.03)	
Immunosuppression (*n*; %)	15 (9.04)	25 (21.55)	<0.0001
**Preoperative laboratory data**			
Leucocytes (×1000/µL) [median ± SD]	9.9 ± 5.69	10.55 ± 5.3	0.899
CRP (mg/dl) [median ± SD]	2.75 ± 9.52	5.4 ± 10.75	0.062
Hemoglobin (g/dl) [median ± SD]	14.1 ± 2.1	13.05 ± 2.13	<0.0001
Thrombocytes (×1000/µL) [median ± SD]	259.0 ± 125.98	269.5 ± 99.9	0.221
**Number of episodes (*n*; %)**			0.326
1 versus > 1	85/81 (51.20/48.80)	74/42 (63.79/36.21)	0.038
**Modified Hinchey classification (*n*; %)**			0.084
0	4 (2.41)	4 (3.45)	
IA	44 (26.51)	18 (15.52)	
IB	66 (39.76)	39 (33.62)	
II	13 (7.83)	14 (12.07)	
III	34 (20.48)	35 (30.17)	
IV	5 (3.01)	6 (5.17)	
Complicated current episode (*n*; %)	120 (72.29)	96 (82.76)	0.045
Emergency intervention (*n*; %)	52 (31.33)	54 (46.55)	0.012
Percutaneous abscess drainage (*n*; %)	9 (5.42)	8 (6.90)	0.62
Failure of initial medical therapy (*n*; %)	12 (7.23)	16 (13.79)	0.104
Emergency surgery (*n*; %)	45 (27.11)	51 (43.97)	0.004
Duration: current episode to surgery (days) [median ± SD]	10.0 ± 36.55	7.0 ± 21.71	0.009
**Primary surgical approach (*n*; %)**			<0.0001
Laparoscopic	102 (61.45)	38 (32.76)	
Open	64 (38.55)	78 (67.24)	
**Transection approach (*n*; %)**			<0.0001
Hartmann’s resection	37 (22.29)	45 (38.79)	
Primary anastomosis without ostomy	118 (71.08)	56 (48.28)	
Primary anastomosis with protective ostomy	11 (6.63)	15 (12.93)	
Operative time (min) [median ± SD]	250.0 ± 82.51	237.5 ± 83.62	0.339

**Table 5 medicina-59-01083-t005:** Multivariate logistic regression analysis of predictors for an eventful postoperative course.

Selected Variables	OR	95% CI	*p*-Value
Leucocytes	0.961	0.913–1.012	0.105
Hemoglobin	0.873	0.768–0.991	0.042
Diabetes			0.491
No	Reference		
Yes	2.495	0.858–7.253	
ASA classification			0.040
1	Reference		
2	1.520	0.626–3.688	
3	2.503	0.942–6.650	
4	7.200	1.441–35.975	
Duration: current episode to surgery	0.991	0.981–1.002	0.116
Primary surgical approach			0.014
Laparoscopic	Reference		
Open	2.897	1.431–5.862	
Operative time	1.003	1.0–1.007	0.049

**Table 6 medicina-59-01083-t006:** Length of postoperative hospital stay in relation to patient, disease, and treatment parameters.

	Total Patients *n* = 282	Hospital Stay (days) [Median ± SD]	*p*-Value
**Sex (*n*; %)**			0.252
Male	136 (48.23)	11.0 ± 18.5	
Female	146 (51.77)	12.0 ± 12.89	
**Diabetes (*n*; %)**			0.019
Yes	20 (7.09)	15.5 ± 13.05	
No	262 (92.91)	11.0 ± 15.58	
**Arterial hypertension (*n*; %)**			<0.0001
Yes	135 (47.87)	12.0 ± 19.04	
No	147 (52.13)	10.0 ± 10.19	
**Chronic renal insufficiency (*n*; %)**			0.014
Yes	32 (11.35)	14.0 ± 28.8	
No	250 (88.65)	11.0 ± 13.12	
**ASA classification (*n*; %)**			<0.0001
1	35 (12.41)	8.0 ± 3.33	
2	118 (41.84)	10.5 ± 11.21	
3	74 (26.24)	14.0 ± 20.33	
4	10 (3.55)	19.5 ± 29.66	
NA	45 (15.96)	NA	
**Immunosuppression (*n*; %)**			<0.0001
Yes	40 (14.18)	14.5 ± 30.06	
No	242 (85.82)	11.0 ± 11.16	
**Number of episodes (*n*; %)**			<0.0001
1	159 (56.38)	13.0 ± 15.1	
>1	123 (43.62)	10.0 ± 16.6	
**Modified Hinchey classification (*n*; %)**			0.001
0	8 (2.84)	9.5 ± 8.86	
IA	62 (21.99)	9.0 ± 6.36	
IB	105 (37.23)	10.0 ± 12.99	
II	27 (9.57)	11.0 ± 10.02	
III	69 (24.47)	14.0 ± 24.28	
IV	11 (3.90)	13.0 ± 14.29	
**Complicated current episode (*n*; %)**			0.003
Yes	216 (76.60)	12.0 ± 17.42	
No	66 (23.40)	9.0 ± 6.71	
**Emergency intervention (*n*; %)**			<0.0001
Yes	106 (37.59)	14.0 ± 22.02	
No	176 (62.41)	10.0 ± 8.63	
**Percutaneous abscess drainage (*n*; %)**			0.007
Yes	17 (6.03)	16.0 ± 21.12	
No	265 (93.97)	11.0 ± 15.33	
**Failure of initial medical therapy (*n*; %)**			0.004
Yes	28 (9.93)	17.0 ± 13.03	
No	254 (90.07)	11.0 ± 16.04	
**Emergency surgery (*n*; %)**			<0.0001
Yes	96 (34.04)	14.0 ± 21.69	
No	186 (65.96)	10.0 ± 10.55	
**Primary surgical approach (*n*; %)**			<0.0001
Laparoscopic	140 (49.65)	9.0 ± 9.8	
Open	142 (50.35)	14.0 ± 19.28	
**Transection approach (*n*; %)**			<0.0001
Hartmann’s resection	82 (29.08)	14.5 ± 23.15	
Primary anastomosis without ostomy	174 (61.70)	10.0 ± 9.35	
Primary anastomosis with protective ostomy	26 (9.22)	15.5 ± 13.79	

**Table 7 medicina-59-01083-t007:** Multivariate linear regression analysis of predictors for length of hospital stay.

Selected Variables	Coefficient (B)	SE	*p*-Value
Hemoglobin	−1.152	0.485	0.018
ASA classification	3.610	1.154	0.002
1	Reference		
2	1.308	2.731	0.632
3	3.546	3.388	0.296
4	11.887	5.111	0.021
Immunosuppression	8.646	3.336	0.010
Emergency intervention	10.689	4.693	0.024
Operative time	0.031	0.010	0.010

## Data Availability

The data presented in this study are available upon request from the corresponding author.

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
