# Peer review of "A New Nomogram-Based Prediction Model for Postoperative Outcome after Sigmoid Resection for Diverticular Disease"

_medicina, 2023, doi:10.3390/medicina59061083_

Round 1
Reviewer 1 Report
A very interesting focus that has yielded some pivotal predictive factors which can improve risk stratification in patients with diverticular disease. My comments are as follows:
1. In the methods and materials section, data was collected from patients who were admitted to the department of surgery. It would be also relevant to report patients who presented to the Emergency Department and were not referred to surgery despite having an indication for further surgery.
2. A strong statistical analysis section indeed. However, can you also please provide details pertaining to sample size calculation to ensure that the readers are aware that the study is adequately powered?
3. In the statistical analysis section, the following is mentioned: "clinical selection model was constructed as recently described". Can you please elaborate more here?
4. In the total co-morbidities reported for the cohort, was any Inflammatory bowel disease noted?
5. Likewise, can you also please report relevant anti-inflammatory or immunomodulating medications the patients were on?
6. How do you propose utilizing WBC count in this predictive model in a patient who is on immunosuppressive therapy chronically?
7. Mean Hemoglobin levels are reported. Can you please also report how many of these patients had prior anemia that could alter the hemoglobin levels? Additionally, also if any patients were receiving iron replacement therapy should also be adjusted for/reported.
8. In patients with chronic renal insufficiency, we know that elevated BUN tends to induce a qualitative defect in patients. Therefore, would also suggest including BUN and Creatinine levels in your outcomes analysis for renal disease patients.
9. Did any of the patients have recent bowel surgeries for indications other than the diverticular disease? This is unclear.
10. Did any of the patients have recent episodes of infectious colitis where they required antibiotics/antivirals etc.? Please clarify and provide information in the text where relevant.
Reviewer 2 Report
Very important topic with appropriate way presented.
Study conclusions and main points can be significant in routine practice.
This kind of prediction models can facilitate our daily work with better understanding of risk factors.
